# Burden of rodent-borne viruses in rodents and zoonotic risk in humans in Cambodia

Julia Guillebaud,[1,2] Janin Nouhin,[1] Vibol Hul,[1] Thavry Hoem,[1] Oudamdaniel Yanneth,[1] Mala Sim,[1] Limmey Khun,[1] Phalla Y,[1] Leangyi Heng,[1] Sreymom Ken,[1] Leakhena Pum,[1] Reaksa Lim,[1] Channa Meng,[3] Kimtuo Chhel,[1] Sithun Nuon,[1] Sreyleak Hoem,[1] Kunthy Nguon,[4] Malen Chan,[4] Sowath Ly,[4] Erik A. Karlsson,[1] Jean-Marc Reynes,[5] Anavaj Sakunthabhai,[6] Philippe Dussart,[1] Veasna Duong[1]

**ABSTRACT**  Rodent-borne viruses, including orthohantaviruses, mammarenaviruses, and rat hepatitis virus (HEV-C), pose significant health threats to humans, causing severe diseases such as hemorrhagic fevers, respiratory illness, and hepatitis. In Cambodia, data on these viruses remain limited, and their burdens on human health are unknown. This study investigated the presence of these viruses in rodents and assessed potential human exposure across diverse environmental and socio-economic contexts in Cambodia. The study was conducted in urban, semi-urban, and rural areas of Cambodia during the rainy season of 2020 and dry season of 2022. Rodents were screened for viruses using RT-PCR. Human serum samples from the same sites were tested for IgG antibodies using ELISA. Factors associated with virus spillover into humans were analyzed. Among 750 rodents, 9.7% carried at least one virus: 5.2% mammarenavirus, 3.3% orthohantavirus, and 1.9% HEV-C. Infection rates were highest in urban (14.5%), followed by semi-urban (11.9%) and rural (2.1%) interfaces. Mammarenavirus was more prevalent in the rainy season, while orthohantavirus and HEV-C remained consistent across seasons. In humans, seroprevalence was 12.7% for mammarenavirus, 10.0% for orthohantavirus, and 24.2% for HEV. Higher mammarenavirus seroprevalence was associated with urban residency. Orthohantavirus seroprevalence was associated with urban residency, acute hepatitis history, and flood-prone living areas. HEV seroprevalence increased with urban residency, increasing age, and medical condition history. Our findings highlighted the need for rodent control, improved market infrastructure, enhanced waste management, and public awareness of hygiene practices and zoonotic risks, especially in urban high-risk areas.

**IMPORTANCE**  Rodents can carry viruses that may spread to humans, sometimes causing serious diseases. However, little was known about the presence of these viruses in Cambodia or their potential impact on human health. This study investigated rodent populations across urban, semi-urban, and rural areas and tested both rodents and humans for three key viruses: arenavirus, hantavirus, and hepatitis E virus. The findings confirm the presence of these viruses in rodents and indicate human exposure, particularly in urban areas. Factors such as urban residency and living in flood-prone areas were associated with an increased risk of exposure. These results emphasize the need for improved rodent control, waste management, and public awareness of zoonotic disease risks. A better understanding of virus transmission dynamics will help guide health officials in developing effective strategies to prevent infections and protect communities.

**KEYWORDS**  rodent-borne viruses, arenavirus, hantavirus, rat hepatitis E virus, Cambodia

Address correspondence to Janin Nouhin, njanin@pasteur-kh.org.

Julia Guillebaud, Janin Nouhin, and Vibol Hul contributed equally to this article. The author order was determined based on their contributions to the article.

Philippe Dussart and Veasna Duong contributed equally to this article.

The authors declare no conflict of interest.

See the funding table on p. 14.

Rodents are reservoirs for a wide range of zoonotic viruses, many of which cause severe disease in humans. Nevertheless, rodent-borne infections remain understudied and neglected in several regions. In Southeast Asia, rapid urban expansion, rice-based agriculture, and extensive human–animal interfaces create conditions that facilitate rodent–human contact and viral spillover. Cambodia, located within the Greater Mekong Subregion, contains diverse ecosystems—including tropical rainforests, wetlands, and rice fields—that support a rich biodiversity of rodent species. Over the past two decades, the country has undergone rapid economic development accompanied by accelerating urbanization and significant environmental changes (1). Such expansion into biodiverse habitats increases the potential for zoonotic spillover. Understanding the burden of these viruses in this setting is essential for anticipating outbreaks and informing targeted surveillance and prevention strategies.

Among the most consequential rodent-borne viruses are orthohantaviruses, mammarenaviruses, and rodent-borne hepatitis E virus (HEV-C). Orthohantaviruses (family *Hantaviridae*) are negative-sense RNA viruses that cause hemorrhagic fever with renal syndrome and hantavirus pulmonary syndrome, with reported case-fatality rates ranging from 1% to more than 30%. Mammarenaviruses (family *Arenaviridae*), best known for Lassa virus and related pathogens, also circulate in Asia, where novel species continue to be discovered. HEV-C, a divergent member of the *Hepeviridae* family maintained in rodents, has emerged as a zoonotic cause of hepatitis, particularly in immunocompromised individuals. These viruses exhibit varying infection dynamics within their natural reservoirs. Orthohantaviruses, such as Seoul orthohantavirus (SEOV), generally cause persistent, asymptomatic infections in rodents, with viral shedding occurring over extended periods (2–4). Similarly, mammarenaviruses typically establish chronic infections in their rodent hosts. In contrast, rat hepatitis E virus (HEV-C) infections are usually acute but can be detected in multiple organs for a limited period. Collectively, these viruses pose significant public health threats due to their potential to cause severe diseases ranging from hepatitis to respiratory and hemorrhagic fevers (5–8). Human infection typically results from direct or indirect contact with infected rodents, their excreta, or contaminated environmental substrates (7–9).

Evidence of orthohantaviruses in Cambodia has been documented in rodents since 1998, with detections reported from the capital city, Phnom Penh, as well as the southern and eastern provinces of the country (10). Serological surveys have also revealed recent infection: 71 out of 459 individuals (15.5%) demonstrated a fourfold rise in IgG titer or seroconversion (11). More recently, the Cardamones variant of Wenzhou virus, which is a member of the mammarenavirus genus, has been identified in brown rats (*Rattus norvegicus*) and Pacific rats (*Rattus exulans*), and human infection has been reported in febrile patients with respiratory symptoms (12). Information on HEV-C is limited in Cambodia. Although not extensively characterized, related strains have been detected in rodents across Southeast Asia, suggesting a plausible regional presence. A virome analysis of rodent lungs collected between 2006 and 2018 identified partial HEV-C genome sequences in Cambodian samples (13). HEV-C has also been reported in samples collected from neighboring countries—including Thailand, Laos PDR, and Vietnam—as well as in China, Indonesia, and Japan (13–17).

Rodent ecology is a contributing factor that shapes the risk of zoonotic spillover. Cambodia hosts a diverse commensal and peri-domestic rodent species: the Asian house shrew (*Suncus murinus*), the Pacific rat (*Rattus exulans*), the brown rat (*Rattus norvegicus*), and the black rat (*Rattus rattus*) are among the most commonly trapped. Their relative abundance varies across environments: *S. murinus* dominates urban habitats, *R. exulans* is widespread in rural and agricultural areas, and *R. norvegicus* and *R. rattus* thrive in dense settlements and along waterways. These ecological patterns, combined with extensive rice agriculture and seasonal flooding, create multiple opportunities for human exposure (10, 12, 13).

Despite these indicators, comprehensive assessment of rodent-borne pathogens across diverse human–animal interfaces in Cambodia is rare, leaving the true prevalence,

genetic diversity, and geographic distribution of mammarenaviruses, orthohantaviruses, and HEV-C largely unknown. Establishing a baseline for these viruses is critical for risk assessment, diagnostic preparedness, and the development of targeted public health interventions.

Here, we present a comprehensive investigation of mammarenaviruses, orthohantaviruses, and HEV-C in small mammals, assessing evidence of human exposure across different environmental and socio-economic contexts in Cambodia. The investigation focused on the capital city, where high rodent–human interaction is anticipated, leading to an elevated risk of transmission. Subsequently, we extended our study to include interfaces representing Cambodia's diverse landscapes, including semi-urban and rural areas.

## MATERIALS AND METHODS

### Study timing and locations

This descriptive study was conducted across three distinct interfaces in Cambodia—urban, semi-urban, and rural—during the 2020 rainy season and 2022 dry season (Fig. 1A). Study sites were selected to represent an urbanization gradient, reflecting different levels of human development and interaction with the environment. Phnom Penh, the capital city, was classified as the urban interface due to its dense population of approximately 2.1 million inhabitants. Five major markets—Phsar Chas, Phsar Kandal, Orussey, Central, and Chbar Ampov—were chosen based on their size and location, prioritizing sites with a high likelihood of rodent–human interactions and the potential for the widespread distribution of contamination within the surrounding neighborhoods. Preah Sihanouk province, located on the western coast, was selected as a semi-urban interface, representing a region undergoing rapid urbanization, transitioning from the original hills and rice fields to a more developed landscape. Three villages—Veal Thum, Veal Meas, and Boeng Vaeng—were selected. Kampong Cham province, known for its predominantly agrarian landscape and extensive rice fields, served as the rural interface, with the villages of Roung Kou, Krasang Pul, and Toul Ampil selected.

### Sample and data collection

At each site, samples were collected from rodents as well as humans residing or working in the same areas. Initial sample and data collection occurred from July to September 2020, with a planned follow-up 6 months later. However, travel restrictions and other non-pharmaceutical interventions in response to the coronavirus disease 2019 (COVID-19) pandemic delayed the second sampling until March–April 2022.

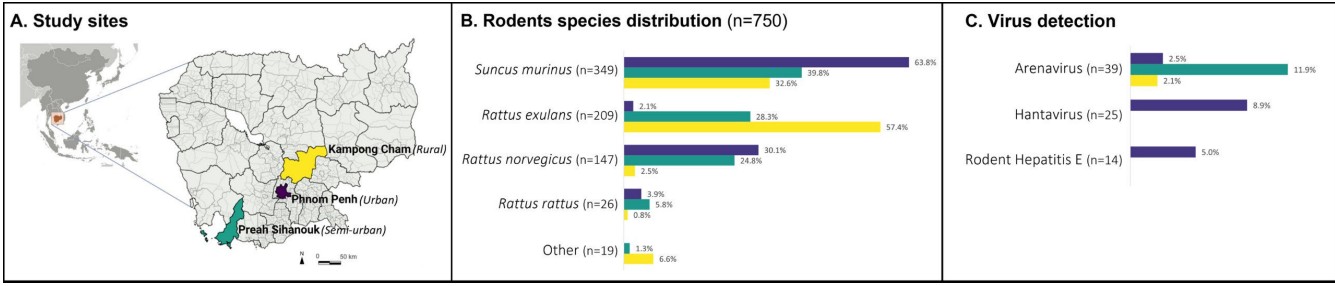

**FIG 1** Study sites, rodent species, and virus prevalence in rodents. (A) Study site: the study was conducted in urban (Phnom Penh, depicted in purple), rural (Kampong Cham province, depicted in yellow), and semi-urban interfaces (Sihanoukville province, depicted in green). (B) Distribution of rodent species: the chart illustrates the overall distribution of rodent species identified in each interface. "Other" category includes *Bandicota savilei* (*n* = 1), *Rattus argentiventer* (*n* = 2), *Mus* sp. (*n* = 9), and *Rattus* sp. (*n* = 7). (C) Virus detection in rodents collected from each interface: the presence of mammarenavirus and orthohantavirus was screened using pooled organ samples (kidney, spleen, liver, and lung) from individual rodent. HEV-C was screened from individual rodent liver samples.

## Rodent sample collection

Rodents were live trapped using locally made traps placed along pathways known to be frequented by rodents. At each site, 50–60 traps were deployed each evening for two to three consecutive nights to capture at least 30 small mammals per session. In the morning, trapped rodents were humanely euthanized following the American Veterinary Medical Association Guidelines (18). Key morphological parameters, including sex, body length, weight, reproductive status, and approximate age (juvenile vs adult), were recorded for species identification. Lung, liver, spleen, and kidney tissues were collected and stored individually in 1 mL of Viral Transport Medium prepared as previously described (19). All samples were preserved on-site in liquid nitrogen and subsequently transferred to the Virology Unit, Institut Pasteur du Cambodge (IPC), where they were stored at −80℃.

## Human sample and data collection

Market workers (≥18 years) from urban interface and villagers (≥2 years) from semi-urban and rural interfaces, residing near rodent trapping sites, were invited to participate in the study. Each participant was requested to provide one blood sample. After obtaining informed consent, trained interviewers conducted face-to-face interviews in Khmer using paper-based questionnaires to collect socio-epidemiological data and information on potential exposure factors. Venous blood (5 mL) was collected in a dry tube, temporarily stored at +4℃ in a cool box, and transported daily to IPC's Virology Unit. Blood samples were centrifuged (10 min, 2,000 rpm). Serum was aliquoted and stored at −80℃ until analysis.

## Assessment of rodent-borne virus infection in animal samples and phylogenetic analysis

Approximately 10 mg (equivalent to 3 mm$^3$) of liver or pooled organs (kidney, spleen, liver, and lung) of each rodent was homogenized using MagNA Lyzer Instrument and bead system (Roche, Basel, Switzerland). Viral RNA was extracted from 200 µL of the supernatant of the homogenized tissue using Direct-Zol RNA MiniPrep Kits (Zymo Research) according to the manufacturer's instructions.

Rodent-borne HEV-C screening was performed on RNA extracted from liver samples using a differential duplex real-time RT-PCR to distinguish it from human-associated hepatitis E virus (HEV-A) (20, 21). Detection of orthohantavirus and mammarenavirus was conducted on RNA extracted from pooled organs using RT-PCR targeting the 412 and 395 bp L segment, respectively, as previously described (22, 23).

All PCR amplified fragments were sent for Sanger sequencing to a commercial sequencing facility (Macrogen, Inc., Seoul, South Korea) using the Big Dye Terminator v3.1 Cycle Sequencing Kit (Applied Biosystems). Chromatograms were sent back electronically to IPC for verification by visual inspection using CLC Genomics Workbench software (CLC bio, Cambridge, MA, USA). Viral sequences were aligned with reference sequences of mammarenavirus, orthohantavirus, and HEV-C retrieved from the GenBank database using MAFFT version 7.490 (24). Phylogenetic trees were constructed using the Neighbor-Joining method with 1,000 bootstrap replicates, based on TN93+G models of nucleotide substitution, as recommended by the "Find Best DNA/Protein Model" tool in the MEGA 11 software (25). Trees were visualized and annotated using FigTree version 1.4.4 (26) and Inkscape 1.2 (https://inkscape.org/). Phylogenetic relationships were assessed using neighbor-joining trees, as the study aimed to determine viral burden rather than perform detailed evolutionary analyses.

## Identification of rodent species

Rodent species were initially identified based on morphological characteristics. Genetic confirmation was attempted for all rodents of each major taxon, using sequences of vertebrate mitochondrial *cytochrome oxidase subunit 1 (CO1)* gene as previously

described (27). Sanger sequencing of the amplified products and visual inspection of chromatograms were conducted as described above. Nucleotide sequences were submitted to NCBI BLAST to determine sequence identity (28).

## Nucleotide sequence accession number

All nucleotide sequences generated in this study were submitted to GenBank and registered under the accession numbers: PV818126–PV818159 (mammarenavirus), PV805852–PV805887, PV845596, PV845597, and PV871972 (orthohantavirus), PV805874–PV805887 (HEV-C), and PV794171–PV794299 (rodent *CO1* gene).

## IgG antibody detection in human serum

We tested participants' sera for IgG antibodies against mammarenavirus and Thailand orthohantaviruses using an in-house enzyme-linked immunosorbent assay (ELISA) (12, 29). Due to the lack of HEV-C-specific ELISA at the time of data analysis, we used a commercial anti-HEV IgG ELISA kit (Beijing Wantai Biological Pharmacy Enterprise, Beijing, China), which primarily detects anti-HEV-A IgG antibody, according to the manufacturer's instructions. Although this assay does not differentiate antibodies against HEV-A and HEV-C, it was the only serological tool available in Cambodia.

Positive samples identified during the primary screening were repeated. The presence of IgG antibodies was considered indicative of previous exposure to the pathogen. However, we acknowledge that HEV seropositivity in our study may limit our ability to directly attribute results to HEV-C. Seroconversion toward a virus was defined as the transition from IgG seronegativity at baseline to seropositivity in the follow-up visit.

## Statistical analysis

Descriptive statistics (Chi-square test and Fisher's exact test) were used to assess the relationships between categorical variables. Exposure to mammarenavirus, orthohantaviruses, and HEV was defined by a seropositive result (positive IgG). Each serological status was analyzed using a generalized linear model with multivariate logistic regression to explore the associations between viral seroprevalence and explanatory variables such as sociodemographic characteristics and potential risk factors. In the final model selection process, while we primarily relied on the stepwise model selection to identify the most parsimonious model based on the Akaike Information Criterion (AIC), we also forced the inclusion of specific variables (e.g., age, sex, and interface) into each model. These variables were retained due to their known biological relevance or potential role as confounders, even though they did not meet the AIC-based selection criteria. A *P*-value of 0.05 was considered statistically significant. Statistical analysis was performed using R software (30).

## RESULTS

### Detection of mammarenavirus, orthohantavirus, and HEV-C in rodents

#### Distribution of rodent species by interface

A total of 750 small mammals, including shrews and rodents, were captured in the two sampling sessions across the three defined interfaces (Fig. 1A). Of these, 647 (86.1%) were adults. The urban interfaces contributed 282 (37.6%), with 114 (40.4%) collected during the rainy season and 168 (59.6%) during the dry season. The semi-urban interface accounted for 226 rodents (30.1%), with 137 (60.6%) collected in the rainy season and 89 (39.4%) in the dry season. In the rural interface, 242 rodents (32.3%) were captured, with 85 (35.1%) collected during the rainy season and 157 (64.9%) in the dry season.

Across all interfaces, the most frequently captured species was *Suncus murinus* (Asian house shrew: 46.5%; 349/750), followed by *Rattus exulans* (Pacific rat: 27.9%; 209/750), *Rattus norvegicus* (brown rat: 19.6%; 147/750), and *Rattus rattus* (black rat: 3.5%; 26/750). Less common species (<2% each) were grouped as "Other" (2.5%; 19/750) and included

*Mus* sp. (unidentified mouse species: 1.2%; 9/750), *Rattus* sp. (unidentified rat species: 0.9%; 7/750), *Rattus argentiventer* (rice field rat: 0.3%; 2/750), and *Bandicota savilei* (Savile's bandicoot rat: 0.1%; 1/750).

Animal species distribution varied significantly across the interfaces ($P < 0.001$) (Fig. 1B). *S. murinus* dominated the urban interface (63.8%; 180/282), followed by *R. norvegicus* (30.1%; 85/282). The semi-urban interface showed a more balanced distribution, with *S. murinus*, *R. exulans*, and *R. norvegicus* accounting for 39.8% (90/226), 28.3% (64/226), and 24.7% (56/226) of collected rodents, respectively. In the rural interface, *R. exulans* was the predominant species (57.4%, 139/242), followed by *S. murinus* (32.6%, 79/242).

### Virus prevalence across interfaces and phylogenetic analysis

Of the 750 rodents collected, 73 (9.7%) were positive for at least one virus of interest, including 68 (9.1%) infected with a single virus and 5 (0.6%) co-infected with two viruses. Specifically, 39 rodents (5.2%) were positive for mammarenavirus, 25 (3.3%) for orthohantavirus, and 14 (1.9%) for HEV-C. Virus prevalence varied significantly across interfaces ($P < 0.001$), with the highest proportion of infected rodents observed in the urban interface (14.5%, 41/282), followed by the semi-urban (11.9%, 27/226) and rural interfaces (2.1%, 5/242). Mammarenavirus was detected in all three interfaces, with prevalences of 2.5% (7/282) in urban, 11.9% (27/226) in semi-urban, and 2.1% (5/242) in rural areas ($P < 0.001$). In contrast, orthohantavirus and HEV-C were exclusively detected in the urban interface, with prevalences of 8.9% (25/282) and 5.0% (14/282), respectively (Fig. 1C). All five rodents with co-infections were from the urban interface, including four with HEV-C/orthohantavirus co-infection and one with orthohantavirus/mammarenavirus co-infection.

Sequencing was attempted for all positive samples. However, 5 samples could not be successfully sequenced for mammarenavirus, leaving 34 sequences for further analysis. Phylogenetic reconstruction demonstrated that 34 of mammarenavirus sequences obtained from Cambodian rodents clustered within lineage 4, aligning closely with sequences previously reported from the country (Fig. 2A). This suggests a stable circulation of lineage 4 mammarenaviruses across multiple rodent hosts and environmental settings. For orthohantavirus, 24 of the 25 positive samples grouped with SEOV, a globally distributed pathogen commonly associated with *R. norvegicus* in urban environments. One additional sequence clustered with Thottapalayam orthohantavirus, historically linked to the Asian house shrew (*S. murinus*), underscoring potential for cross-species maintenance of this virus in peri-domestic settings (Fig. 2B). Finally, all 14 HEV-C sequences clustered within genotype HEV-C1 (Fig. 2C), with high nucleotide identity to previously reported strains from other countries in the region. This finding indicates a sustained presence of HEV-C1 in Cambodian rodent populations and supports the regional distribution of this genotype across Southeast Asia.

### Seasonal variations in virus prevalence

The infection rate for at least one virus was significantly higher during the rainy season (12.8%, 43/336) versus the dry season (7.2%, 30/414) ($P = 0.01$). The mammarenavirus prevalence was significantly higher during the rainy season, particularly in the semi-urban interface (overall: 7.7% in the rainy season vs 3.1% in the dry season, $P = 0.005$; semi-urban: 16.8% in the rainy season vs 4.5% in the dry season, $P = 0.006$). Detection rates for orthohantavirus and HEV-C were consistent across seasons (Table 1).

### Variations in host species

Among the 39 rodents infected with mammarenavirus, the majority were *R. exulans* (74.4%; 29/39), followed by *R. norvegicus* (17.9%; 7/39) and *R. rattus* (7.7%; 3/39). Of the 25 animals infected with orthohantavirus, 92% (23/25) were *R. norvegicus*, with one case each in *R. rattus* (4%) and *S. murinus* (4%). Both *R. norvegicus* and *R. rattus* were infected

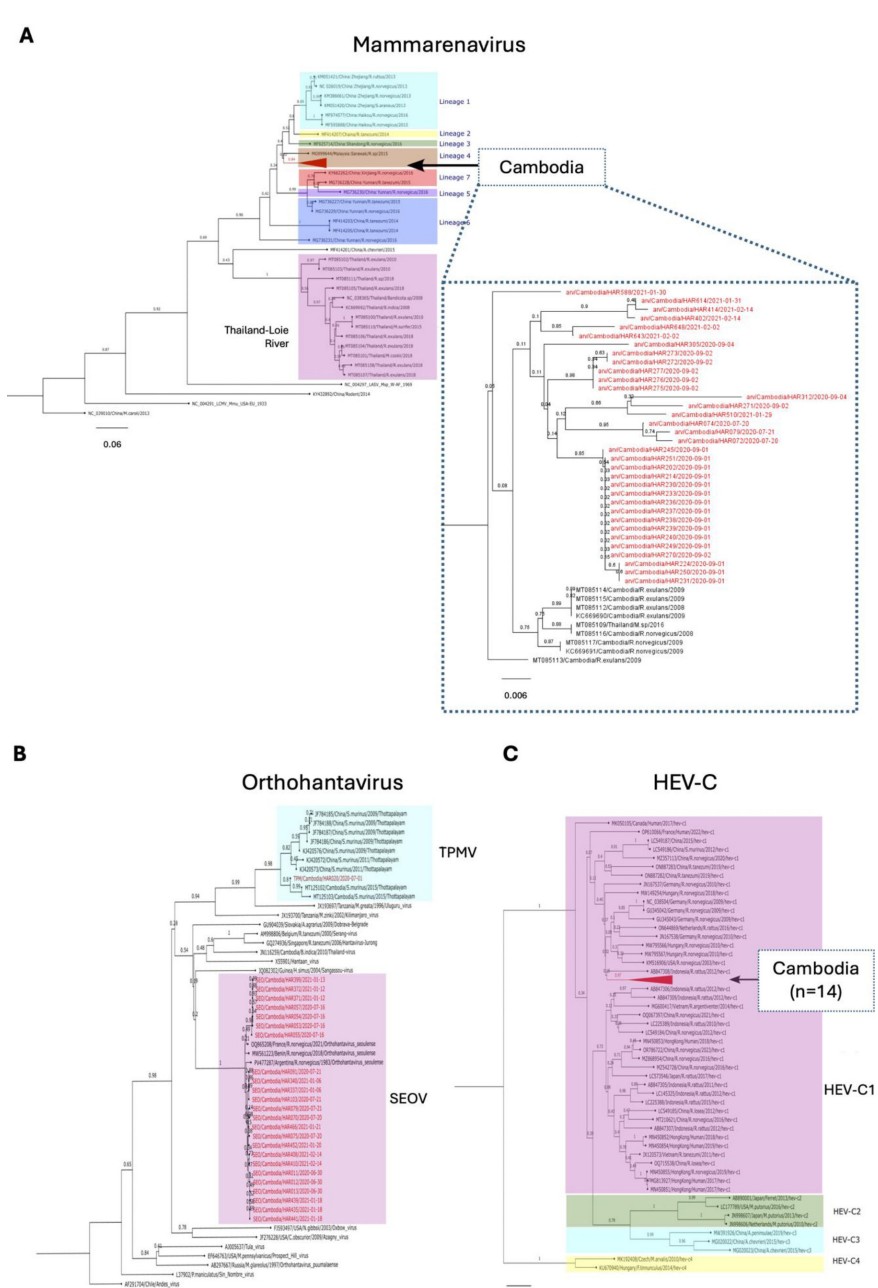

**FIG 2** Phylogenetic trees of mammarenavirus, orthohantavirus, and HEV-C. Phylogenetic trees were inferred using the Neighbor-Joining method with 1,000 bootstrap replicates, based on TN93+G models of nucleotide substitution, as recommended by the "Find Best DNA/Protein Model" tool in the MEGA 11 software. Trees were visualized and annotated using FigTree version 1.4.4 and Inkscape 1.2. (A) Mammarenavirus phylogeny: 34 sequences from the present study (indicated in red) and GenBank reference sequences (indicated in various colors). (B) Orthohanta phylogeny: 25 sequences from the present study (indicated in red) and GenBank reference sequences (indicated in black). (C) HEV-C phylogeny: 14 sequences from the present study (indicated in red triangle) and GenBank reference sequences (indicated in various colors). TN93+G, Tamura-Nei and Gamma distribution; TPMV, Thottapalayam orthohantavirus; SEOV, Seoul orthohantavirus.

with SEOV, whereas *S. murinus* was infected with Thottapalayam orthohantavirus. All 14 rodents infected with HEV-C were *R. norvegicus*. All four rodents co-infected with

**TABLE 1** Detection rate of mammarenavirus, orthohantavirus, and HEV-C in rodents per interface and season[a]

| | Overall | | | Urban | | | Semi-urban | | | Rural | | |
|---|---|---|---|---|---|---|---|---|---|---|---|---|
| | Rainy season | Dry season | P value[b] | Rainy season | Dry season | P value[b] | Rainy season | Dry season | P value[b] | Rainy season | Dry season | P value[b] |
| | (n = 336) | (n = 414) | | (n = 114) | (n = 168) | | (n = 137) | (n = 89) | | (n = 85) | (n = 157) | |
| Mammarenavirus | 26 (7.7%) | 13 (3.1%) | 0.005 | 3 (2.6%) | 4 (2.4%) | 1.000 | 23 (16.8%) | 4 (4.5%) | 0.006 | 0 (0%) | 5 (3.2%) | 0.165 |
| Orthohantavirus | 13 (3.9%) | 12 (2.9%) | 0.46 | 13 (11.4%) | 12 (7.1%) | 0.217 | 0 (0%) | 0 (0%) | –[c] | 0 (0%) | 0 (0%) | – |
| HEV-C | 8 (2.4%) | 6 (1.4%) | 0.35 | 8 (7.0%) | 6 (3.6%) | 0.219 | 0 (0%) | 0 (0%) | – | 0 (0%) | 0 (0%) | – |

[a]Data are n (%).
[b]P value was calculated using a Chi-squared or Fisher's exact tests.
[c]"–" indicates not applicable.

HEV-C/orthohantavirus were *R. norvegicus*, while the rodent co-infected with orthohantavirus/mammarenavirus was *R. rattus*.

## Evidence of spillover of rodent-borne viruses in humans

During the initial visit (July–September 2020), 788 human participants were enrolled across the three interfaces. Of these, 304 (38.6%) participants were from urban areas, 288 (36.5%) from semi-urban areas, and 196 (24.9%) from rural areas. Females accounted for 64.5% (508/788) of the participants, with a median age of 38 years (interquartile range, IQR: 13–54). Table 2 provides a summary of sociodemographic characteristics and potential risk factors associated with each interface type.

Serological evidence of exposure to rodent-borne viruses was observed in the study cohort, with seroprevalence of 12.7% (100/788) for mammarenavirus, 10.0% (79/788) for orthohantavirus, and 24.2% (191/788) for HEV among the total cohort. Due to the limited specificity of the Wantai ELISA assay, HEV seropositivity likely reflects prior exposure to HEV-A, rather than HEV-C, and therefore cannot be interpreted as evidence of zoonotic HEV-C transmission.

Analysis by interface revealed notable differences in seroprevalence, with urban areas exhibiting the highest rates compared to semi-urban and rural areas. For mammarenavirus, 21.7% (66/304) of individuals were seropositive in urban areas, compared to 5.6% and 9.2% in semi-urban and rural areas, respectively (*P* < 0.001). For orthohantavirus, 13.2% (40/304) were positive in urban areas, compared to 9.4% and 6.1% in semi-urban and rural areas, respectively (*P* = 0.03). For HEV, 41.1% (125/304) were positive in urban areas, compared to 14.9% and 11.7% in semi-urban and rural areas, respectively (*P* < 0.001).

## Factors associated with rodent-borne virus spillover into humans

All collected variables, including demographics, residence characteristics, education, occupation, medical history, animal contact, household facility, food-storage practices, and history of flooding, were included in univariate analysis. Factors showing significant associations (interface, age, education, medical history, flooding history, proximity to a swine farm, contact with poultry, alcohol consumption, and consumption of raw shellfish) were entered into multivariate models. Multivariate analysis identified several factors significantly associated with higher seroprevalence of rodent-borne viruses in humans (Fig. 3). Mammarenavirus IgG seroprevalence was associated with urban residency (adjusted odds ratio [ORa] = 2.5, 95% CI: 1.3–4.9, *P* = 0.01) and lower education (ORa = 2.2, 95% CI: 1.1–4.8, *P* = 0.03). Orthohantavirus IgG seroprevalence was associated with urban residency (ORa = 2.3, 95% CI: 1.1–5.3, *P* = 0.03), a history of acute hepatitis (ORa = 5.3, 95% CI: 1.5–16.3, *P* = 0.01), and living in flood-prone areas (ORa = 2.0, 95% CI: 1.0–3.9, *P* = 0.04). HEV IgG seroprevalence was linked to urban residency (ORa = 2.0, 95% CI: 1.1–3.6, *P* = 0.02), increasing age (18–35 years: ORa = 9.3, 95% CI: 2.7–43.3, *P* < 0.001; 36–50 years: ORa = 37.6, 95% CI: 12.3–164, *P* < 0.001; ≥50 years: ORa = 75.9, 95% CI: 25.5–329, *P* < 0.001), and a history of medical conditions (ORa = 1.6, 95% CI: 1.0–2.4, *P* = 0.04). Interestingly, a higher education level (above primary school) was associated with lower HEV IgG seroprevalence (ORa = 0.5, 95% CI: 0.3–0.9, *P* = 0.04).

**TABLE 2** Baseline characteristics[a]

| | Overall (*n* = 788) | Urban (*n* = 304) | Semi-urban (*n* = 288) | Rural (*n* = 196) | *P* value[b] |
|---|---|---|---|---|---|
| Gender | | | | | 0.006 |
| Female | 508 (64.5%) | 213 (70.1%) | 166 (57.6%) | 129 (65.8%) | |
| Male | 280 (35.5%) | 91 (29.9%) | 122 (42.4%) | 67 (34.2%) | |
| Age (years) | 38 (13–54) | 50 (40–60) | 17 (10–47) | 15 (9–41) | <0.001 |
| Age group (years) | | | | | <0.001 |
| <18 | 255 (32.4%) | 105 (53.6%) | 149 (51.7%) | 1 (0.3%) | |
| 18–35 | 112 (14.2%) | 27 (13.8%) | 33 (11.5%) | 52 (17.1%) | |
| 36–50 | 174 (22.1%) | 27 (13.8%) | 46 (16.0%) | 101 (33.2%) | |
| ≥50 | 247 (31.3%) | 37 (18.9%) | 60 (20.8%) | 150 (49.3%) | |
| Education | | | | | <0.001 |
| No school | 127 (16.1%) | 48 (15.8%) | 48 (16.7%) | 31 (15.8%) | |
| Primary | 405 (51.4%) | 129 (42.4%) | 156 (54.2%) | 120 (61.2%) | |
| Secondary and higher | 256 (32.5%) | 127 (41.8%) | 84 (29.2%) | 45 (23.0%) | |
| Occupation | | | | | <0.001 |
| Agricultural worker | 42 (5.3%) | 0 (0.0%) | 6 (2.1%) | 36 (18.4%) | |
| Seller | 311 (39.5%) | 245 (80.6%) | 49 (17.0%) | 17 (8.7%) | |
| Other | 435 (55.2%) | 59 (19.4%) | 233 (80.9%) | 143 (73.0%) | |
| Residency | | | | | |
| Less than 1 year | 14 (1.8%) | 4 (1.3%) | 3 (1.0%) | 7 (3.6%) | |
| More than 1 year | 212 (26.9%) | 146 (48.0%) | 29 (10.1%) | 37 (18.9%) | |
| Entire life | 562 (71.3%) | 154 (50.7%) | 256 (88.9%) | 152 (77.6%) | |
| Any medical condition | | | | | |
| Previous | 146 (18.5%) | 84 (27.6%) | 39 (13.5%) | 23 (11.7%) | <0.001 |
| Chronic | 49 (6.2%) | 35 (11.5%) | 7 (2.4%) | 7 (3.6%) | <0.001 |
| Alcohol consumption | 599 (76.0%) | 198 (65.1%) | 251 (87.2%) | 150 (76.5%) | <0.001 |
| Any acute hepatitis history | 15 (1.9%) | 6 (2.0%) | 9 (3.1%) | 0 (0.0%) | 0.03 |
| Living conditions | | | | | |
| Flooded area | 118 (15.0%) | 19 (6.2%) | 94 (32.6%) | 5 (2.6%) | <0.001 |
| Swine farm nearby | 116 (14.7%) | 44 (14.5%) | 30 (10.4%) | 42 (21.4%) | 0.004 |
| Slaughterhouse nearby | 66 (8.4%) | 58 (19.1%) | 7 (2.4%) | 1 (0.5%) | <0.001 |
| Any wild animal hunting experience | 87 (11.0%) | 15 (4.9%) | 52 (18.1%) | 20 (10.2%) | <0.001 |
| Dietary habits | | | | | |
| Raw pork meat | 64 (8.1%) | 21 (6.9%) | 34 (11.8%) | 9 (4.6%) | 0.011 |
| Pork sausage | 148 (18.8%) | 50 (16.4%) | 63 (21.9%) | 35 (17.9%) | 0.20 |
| Raw shells | 311 (39.5%) | 123 (40.5%) | 102 (35.4%) | 86 (43.9%) | 0.20 |
| Rodent meat | 13 (1.6%) | 5 (1.6%) | 4 (1.4%) | 4 (2.0%) | 0.88 |
| Experience of direct contact with pork | 514 (65.2%) | 229 (75.3%) | 162 (56.2%) | 123 (62.8%) | <0.001 |
| Experience of direct contact with poultry | 502 (63.7%) | 221 (72.7%) | 163 (56.6%) | 118 (60.2%) | <0.001 |
| Experience of contacts with rodents | | | | | |
| Indirect contact | 721 (91.5%) | 296 (97.4%) | 252 (87.5%) | 173 (88.3%) | >0.9 |
| Direct contact | 135 (17.1%) | 53 (17.4%) | 58 (20.1%) | 24 (12.2%) | 0.076 |
| Got bitten by a rodent | 45 (5.7%) | 28 (9.2%) | 11 (3.8%) | 6 (3.1%) | 0.003 |

[a]Data are *n* (%) and median (IQR).
[b]*P* value was calculated using a Chi-squared test.

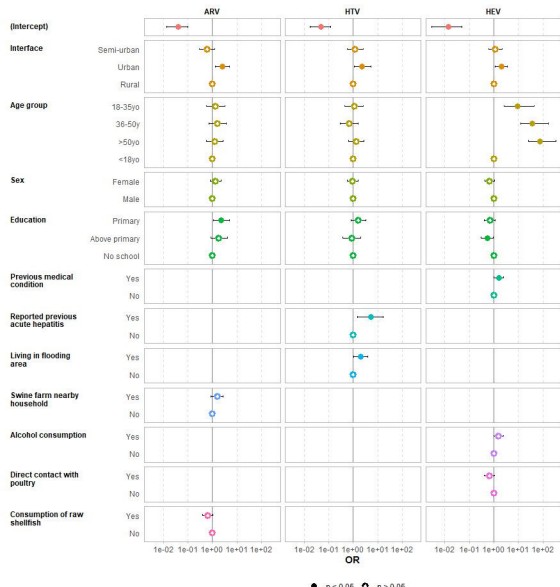

**FIG 3** Forest plot showing factors associated with exposure to mammarenavirus (ARV), orthohantavirus (HTV), and hepatitis E virus (HEV) in human population. OR, odds ratio; CI, confidence interval.

## Follow-up visits and seroconversion rates

In 2022, 555 (70.4%) of the 788 included participants were able to participate in the follow-up visit. Of these, 39.1% (217/555) of participants were from urban, 36.6% (203/555) from semi-urban, and 24.3% (135/555) from rural interfaces (Table 3). Among participants who were seropositive at inclusion, 36.4% (28/77) maintained their seropositive status for anti-mammarenavirus IgG, 89.5% (51/57) for anti-orthohantavirus IgG, and 96.2% (125/130) for anti-HEV IgG.

Seroconversion was observed in 17.6% (84/478) of participants for mammarenavirus, 1.2% (6/498) for orthohantavirus, and 2.1% (9/425) for HEV. For mammarenavirus, the majority of seroconversions occurred in participants from rural interfaces (63.1%), followed by semi-urban (25.0%) and urban (11.9%) interfaces ($P < 0.001$). Participants who had lived in the study area their entire lives showed a significantly higher seroconversion rate (72.6%, 61/84) than those residing there for less than a year (4.8%, 4/84) or more than a year (22.6%, 19/84) ($P = 0.02$).

**TABLE 3** Follow-up serological results of participants for each tested virus and per interface[a]

|  | Overall | Urban | Semi-urban | Rural |
|---|---|---|---|---|
| Inclusion, n (%) | 788 (100%) | 304 (38.6%) | 288 (36.5%) | 196 (24.9%) |
| Follow-up, n (%) | 555 (70.4%) | 217 (71.4%) | 203 (70.5%) | 135 (68.9%) |
| Mammarenavirus |  |  |  |  |
| IgG seropositive at inclusion | 77 (13.9%) | 52 (24.0%) | 11 (5.4%) | 14 (10.4%) |
| Participants remain seropositive | 28 (36.4%) | 11 (21.2%) | 6 (54.5%) | 11 (78.6%) |
| IgG seroconversion[b] | 84 (17.6%) | 10 (6.1%) | 21 (10.9%) | 53 (43.8%) |
| Orthohantavirus |  |  |  |  |
| IgG seropositive at inclusion | 57 (10.3%) | 29 (13.4%) | 18 (8.9%) | 10 (7.4%) |
| Remain seropositive | 51 (89.5%) | 27 (93.1%) | 17 (94.4%) | 7 (70.0%) |
| IgG seroconversion[b] | 6 (1.2%) | 2 (1.1%) | 1 (0.5%) | 3 (2.4%) |
| HEV |  |  |  |  |
| IgG seropositive at inclusion | 130 (23.4%) | 85 (39.2%) | 32 (15.8%) | 13 (9.6%) |
| Remain seropositive | 125 (96.2%) | 82 (96.5%) | 31 (96.9%) | 12 (92.3%) |
| IgG seroconversion[b] | 9 (2.1%) | 6 (4.5%) | 2 (1.2%) | 1 (0.8%) |

[a]Data are n (%).
[b]Participant presenting a seronegative result at inclusion and a seropositive result at follow-up visit.

## DISCUSSION

Rodent-borne viruses represent an important public health concern in Cambodia. The country's tropical climate, widespread agricultural activities, and rapid urbanization expansion create ideal conditions for rodent population growth and increased human–rodent interactions, heightening the risk of zoonotic spillover (31). Inadequate housing structures and limited pest control further amplify exposure risks. Additionally, gaps in healthcare infrastructure and limited awareness of these pathogens contribute to their neglect. Understanding the prevalence and distribution of these viruses is essential for designing targeted interventions to mitigate their impact.

This study assessed the prevalence of mammarenavirus, orthohantavirus, and HEV-C in rodents and evidence of exposure in humans across urban, semi-urban, and rural interfaces during both rainy and dry seasons. All three viruses were detected in Cambodia, with the highest prevalence observed in densely populated urban rodents (2.5% for mammarenavirus, 8.9% for orthohantavirus, and 5.0% for HEV-C), consistent with reports from Vietnam, Indonesia, and China (16, 32–34). Our data show that these viruses are host specific, for example, *Rattus exulans* was the primary reservoir for mammarenavirus, whereas *R. norvegicus* served as the dominant host for both orthohantavirus and HEV-C and accounted for all co-infections. Species-specific traits such as habitat preference, social structure, and population density can influence exposure and intraspecies transmission, while certain viruses may exhibit host-adaptive features enhancing replication in particular rodent species (35, 36). Sampling limitations, including trapping success and habitat accessibility, may also contribute to the observed patterns. Seasonal analysis revealed that mammarenavirus transmission intensified during the rainy season, especially in semi-urban areas, likely explained by ecological factors such as flood-related rodent habitat disruption, increased rodent density, and increased opportunities for human exposure, while no clear seasonal trend was observed for orthohantavirus and HEV-C.

The presence of multiple rodent-borne viruses in urban rodents likely reflects the dense rodent communities in these environments, which facilitate virus transmission. Our urban rodents were trapped mainly in markets, where abundant food, poor waste management, and a lack of rodent control programs promote rodent population growth and interspecies contact. Many of these rodents inhabit sewage systems that flood during the rainy season, forcing them into crowded habitats and further increasing transmission potential. In contrast, rodents from semi-urban and rural interfaces were mainly collected from households unaffected by flooding, where such conditions are less pronounced. Phylogenetic data demonstrate that rodent-borne viruses detected in Cambodia predominantly belong to previously recognized lineages, rather than representing novel viral diversity. The predominance of SEOV and HEV-C1 in urban *R. norvegicus* highlights the importance of commensal rodents in maintaining transmission cycles in densely populated environments, whereas the detection of Thottapalayam orthohantavirus in *S. murinus* illustrates the broader host spectrum that may contribute to zoonotic risk (10, 12).

Human serological data mirrored the ecological patterns observed in rodents, confirming ongoing zoonotic exposure. Overall seroprevalences were 12.7% for mammarenavirus, 10.0% for orthohantavirus, and 24.2% for HEV, with a clear urban–rural gradients: urban residents showed the highest rate (22%, 13%, and 41%, respectively) compared with rural residents (9%, 6%, and 12%). Risk factor analysis further highlighted urban residency as a key driver of exposure, consistent with the high virus prevalence in urban rodents, except for mammarenavirus.

The seropositivity of mammarenavirus in humans was unexpectedly high in urban interfaces, despite low prevalence in rodents. This discrepancy could be explained by the intense human–rodent contact in urban markets, where even a small number of infected *R. exulans* can facilitate virus transmission to multiple individuals. The seropositivity of mammarenavirus fluctuated across the three interfaces, likely due to a combination of rodent ecological factors (density and species composition), environmental

pressure (flooding and habitat disturbance), and human socio-behavioral differences (housing quality, food storage, and occupational exposure). In urban setting, dense population of rodents and humans, constant exposure to contaminated food, poorly managed waste, and frequent flooding of sewer systems likely explain the high initial antibody prevalence. However, once a large fraction of the urban population is immuned, new seroconversions may slowdown, creating a plateau effect over time. In contrast, semi-urban settings may have lower rodent densities and moderately improved sanitation, resulting in the lowest antibody prevalence. Rural communities, despite only moderate baseline prevalence, experienced the highest incidence of new infections, probably driven by seasonal surges of *R. exulans* linked to rice harvesting, grain storage, and flood-related rodent displacement, as well as episodic high-contact events such as market visits or agricultural activities.

Orthohantavirus seroprevalence in humans aligns with previous studies in Cambodia (11) but exceeds rates reported in Vietnam, where a survey of healthy donors, febrile patients, and farming communities documented lower prevalence (1.1%–3.7%) (37, 38), highlighting geographic heterogeneity in human exposure. A history of acute hepatitis was significantly associated with orthohantavirus seropositivity, echoing rare reports from Malaysia and Japan. One hypothesis is that orthohantavirus infection may trigger acute exacerbation of autoimmune liver disease, though community-acquired hepatitis could not be excluded. While orthohantavirus infections rarely cause hepatitis without renal manifestation, this association warrants further investigation (39, 40). Living in flood-prone areas was also associated with higher orthohantavirus seroprevalence, likely due to rodent habitat disruption and environmental contamination during floods (41).

Interpretation of the HEV results requires caution, as the Wantai ELISA Kit used cannot differentiate HEV-A from HEV-C (42). Although HEV-C RNA was detected in rodents, human serological findings likely reflect exposure to endemic HEV-A. We therefore cannot infer zoonotic transmission of HEV-C. Future studies using HEV-C-specific antigens or neutralization assays are needed to evaluate potential zoonotic transmission of HEV-C more accurately. Older age was a significant risk factor for higher HEV seroprevalence, as described in previous studies, reflecting cumulative lifetime exposure, while higher education levels appeared to be protective (43, 44). Nevertheless, the association between seropositivity and education level did not show a clear dose–response relationship, and a borderline *P*-value suggests caution; other confounding factors, such as socioeconomic status, occupational exposure, and hygiene practices, may influence this association.

We observed that a substantial proportion of participants maintained seropositivity over the 2-year follow-up—36.4% for mammarenavirus, 89.5% for orthohantavirus, and 96.2% for HEV. This pattern suggests that antibodies to orthohantavirus and HEV are comparatively long-lasting, whereas mammarenavirus antibodies may wane more rapidly. These data have implications for interpreting seroprevalence as a marker of cumulative exposure and for designing future longitudinal surveillance studies.

Several limitations should be noted in this study. First, the delay in follow-up due to the COVID-19 pandemic extended the timeframe between visits, limiting our ability to evaluate short-term incidence. Second, seroprevalence estimates may be overestimated due to potential cross-reactivity with other closely related viruses and require further confirmatory testing. Third, the ELISA kit used for HEV-C serological assay cannot differentiate HEV-A and HEV-C, reducing the specificity in serological findings. Further studies will focus on developing and employing more robust diagnostic tools and methodologies to reduce these issues.

Overall, these findings underscore the interplay between rodent ecology, human behavior, and environmental factors in shaping zoonotic risk. They highlight the critical need for integrated interventions, including effective rodent control, improved market infrastructure, enhanced waste management, public-awareness campaigns on hygiene and zoonotic hazards, particularly in urban and high-risk areas. Evidence of virus exposure in humans underscores the importance of sustained, nationwide surveillance

and the development of HEV-C-specific diagnostics to monitor virus dynamics and assess their clinical impact. Finally, our findings inform medical professionals regarding diseases caused by rodent-borne viruses, enabling better differential diagnoses and improved patient care. Indeed, this study reinforces the value of a One Health approach to understand and mitigate zoonotic spillover while fostering sustainable human–animal coexistence.

## ACKNOWLEDGMENTS

The authors are thankful to all relevant authorities from the Ministry of Agriculture, Forestry and Fisheries of Cambodia for their support in facilitating this work. The authors particularly acknowledge the Department of Wildlife and Biodiversity under the Forestry Administration for their continuous support and participation during field data collection. The authors are grateful to local authorities for facilitating all field work in selected areas and providing valuable human resources. The authors especially express their gratitude to all participants of this study and are grateful to R. Johne (Federal Institute for Risk Assessment, Germany), who generously shared HEV-C positive control as a dried plasmid containing the whole genome of ratHEV strain R63.

The study was supported in part by internal funding of the Institut Pasteur du Cambodge. The study on orthohantavirus was a Centers for Research in Emerging Infectious Diseases (CREID) Pilot Research Program 2021 funded by the National Institute of Allergy and Infectious Diseases of the National Institutes of Health under Award Number 1U01AI151378 through Pasteur International Center for Research in Emerging Infectious Diseases (PICREID). The content is solely the responsibility of the authors and does not necessarily represent the official views of the National Institutes of Health. The funders of the study had no role in study design, data collection, data analysis, data interpretation, or writing of the report.

V.H., J.G., J.N., P.D., and V.D. conceived and designed the study. J.G., T.H., O.Y., and M.S. performed field work to collect samples. J.G., V.H., and S.L. prepared the questionnaire for human epidemiological data. K.N. and M.C. collected human epidemiological data. O.Y., M.S., L.K., L.P., R.L., K.C., S.N., and S.H. carried out molecular assays for virus screening and detection. O.Y., M.S., P.Y., L.H., and S.K. performed serological assays. J.G., V.H., T.H., and J.N. assessed and verified all data for analysis in this study. J.G. and O.Y. performed statistical analysis. J.-M.R. led technological transfer of serological assay to detect the exposure to orthohantavirus. J.G. and J.N. drafted the manuscript. E.A.K., V.D., and A.S. provided critical revision of the manuscript and intellectual input. All authors reviewed and approved the final draft for submission. All authors had full access to all the data in the study and had final responsibility for the decision to submit for publication.

## AUTHOR AFFILIATIONS

[1]Virology Unit, Institut Pasteur du Cambodge, Pasteur Network, Phnom Penh, Cambodia
[2]UMR ASTRE, International Centre of Research in Agriculture for Development (CIRAD), Montpellier, France
[3]Department of Wildlife and Biodiversity, Forestry Administration, Ministry of Agriculture, Forestry and Fisheries, Phnom Penh, Cambodia
[4]Epidemiology and Public Health Unit, Institut Pasteur du Cambodge, Pasteur Network, Phnom Penh, Cambodia
[5]Unité Environnement et Risque Infectieux, Université Paris Cité, Institut Pasteur, Paris, France
[6]Human Genetics Unit, Institut Pasteur, Paris, France

## PRESENT ADDRESS

Philippe Dussart, Institut Pasteur, Paris, France

## AUTHOR ORCIDs

Janin Nouhin http://orcid.org/0000-0003-4985-8377
Erik A. Karlsson http://orcid.org/0000-0001-6004-5671
Veasna Duong http://orcid.org/0000-0003-0353-1678

## FUNDING

| Funder | Grant(s) | Author(s) |
|---|---|---|
| National Institute of Allergy and Infectious Diseases | 1U01AI151378 | Janin Nouhin |

## AUTHOR CONTRIBUTIONS

Julia Guillebaud, Conceptualization, Data curation, Formal analysis, Investigation, Methodology, Validation, Visualization, Writing – original draft, Writing – review and editing | Janin Nouhin, Conceptualization, Data curation, Formal analysis, Funding acquisition, Investigation, Methodology, Project administration, Software, Supervision, Validation, Visualization, Writing – original draft, Writing – review and editing | Vibol Hul, Conceptualization, Data curation, Formal analysis, Investigation, Methodology, Project administration, Supervision, Validation, Visualization, Writing – original draft, Writing – review and editing | Thavry Hoem, Data curation, Formal analysis, Investigation, Methodology, Validation, Visualization, Writing – original draft, Writing – review and editing | Oudamdaniel Yanneth, Conceptualization, Data curation, Formal analysis, Investigation, Validation, Visualization, Writing – original draft, Writing – review and editing | Limmey Khun, Data curation, Formal analysis, Investigation, Writing – review and editing | Phalla Y, Data curation, Formal analysis, Investigation, Validation, Writing – review and editing | Leangyi Heng, Data curation, Formal analysis, Investigation, Validation, Writing – review and editing | Sreymom Ken, Data curation, Formal analysis, Investigation, Supervision, Validation, Writing – review and editing | Leakhena Pum, Data curation, Formal analysis, Investigation, Writing – review and editing | Reaksa Lim, Data curation, Formal analysis, Investigation, Writing – review and editing | Channa Meng, Investigation, Project administration, Resources, Writing – review and editing | Kimtuo Chhel, Investigation, Writing – review and editing | Sithun Nuon, Investigation, Writing – review and editing | Sreyleak Hoem, Investigation, Writing – review and editing | Kunthy Nguon, Data curation, Formal analysis, Investigation, Writing – review and editing | Malen Chan, Data curation, Formal analysis, Investigation, Writing – review and editing | Sowath Ly, Conceptualization, Investigation, Methodology, Project administration, Supervision, Writing – review and editing | Jean-Marc Reynes, Conceptualization, Funding acquisition, Investigation, Methodology, Supervision, Writing – review and editing | Anavaj Sakunthabhai, Conceptualization, Funding acquisition, Investigation, Methodology, Supervision, Writing – review and editing | Philippe Dussart, Conceptualization, Funding acquisition, Investigation, Methodology, Project administration, Supervision, Writing – review and editing | Veasna Duong, Conceptualization, Funding acquisition, Investigation, Methodology, Supervision, Writing – review and editing.

## DATA AVAILABILITY

All nucleotide sequences generated in this study were submitted to GenBank and registered under the accession numbers: PV818126–PV818159 (mammarenavirus), PV805852–PV805887, PV845596, PV845597, and PV871972 (orthohantavirus), PV805874–PV805887 (HEV-C), and PV794171–PV794299 (rodent CO1 gene).

## ETHICS APPROVAL

Study protocols were approved by the Cambodian National Ethics Committee for Health Research (NECHR) (No. 320 NECHR, No. 132 NECHR, No. 215 NECHR, and No. 256 NECHR). Written informed consent for the use of demographic and biological data was obtained

from adult participants or guardians of participants under 18 years old. Animal capture and handling followed American Society of Mammalogists guidelines (18, 45). Statutory study permission was granted by the Forestry Administration of the Cambodian Ministry of Agriculture, Forestry and Fisheries, the national authority responsible for wildlife research.

## ADDITIONAL FILES

The following material is available online.

### Open Peer Review

**PEER REVIEW HISTORY (review-history.pdf).** An accounting of the reviewer comments and feedback.

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
