## [Reviewer comments · Microbiology Spectrum]

Microbiology Spectrum

Burden of rodent-borne viruses in rodents and zoonotic risk in human in Cambodia

Julia Guillebaud, Janin NOUHIN, Vibol Hul, Thavry Hoem, Oudamdaniel Yanneth, Mala Sim, Limmey Khun, Phalla Y, Leangyi Heng, Sreymom Ken, Leakhena Pum, Reaksa Lim, Channa Meng, Kimtuo Chel, Sithun Nuon, Sreyleak Hoem, Kunthy Nguon, Malen Chan, Sowath Ly, Erik Karlsson, Jean-Marc Reynes, Anavaj Sakuntabhai, Philippe Dussart, and Veasna Duong

Corresponding Author(s): Janin NOUHIN, Institut Pasteur du Cambodge

Review Timeline:

Submission Date:	October 1, 2025
Editorial Decision:	October 9, 2025
Revision Received:	October 11, 2025
Accepted:	October 13, 2025

Editor: Peter Pelka

Reviewer(s): The reviewers have opted to remain anonymous.

Transaction Report:

DOI: <https://doi.org/10.1128/spectrum.01394-25>

Re: Spectrum01394-25 (**Burden of rodent-borne viruses in rodents and zoonotic risk in human in Cambodia**)

Dear Dr. Janin NOUHIN:

Thank you for the privilege of reviewing your work. Below you will find my comments and instructions from the Spectrum editorial office.

I am pleased to inform you that your manuscript has been editorially accepted for publication. However, there are a few additional questions in the submission form that need to be answered before the final decision. Once these are completed, please return your submission so that I can move your paper forward to acceptance.

Revision Guidelines

Sincerely,
Peter Pelka
Editor
Microbiology Spectrum

Response to reviewers

Reviewer #1 (Comments for the Author):

This study investigates the presence of three zoonotic viruses, including arenavirus, hantavirus, and HEV-C, in rodent populations in Cambodia, as well as the potential zoonotic risk to humans through seroprevalence analysis. The topic is relevant and important for public health surveillance. However, the manuscript is not well written, and the presentation of the results needs

improvement. Several concerns need to be addressed.

1. A primary concern is the apparent conflation of HEV-C infection in rodents with HEV-A seroprevalence in humans. HEV-C and HEV-A belong to different genera, and the serological assay used likely detects only HEV-A. Therefore, the study does not provide sufficient evidence to support zoonotic transmission of HEV-C in Cambodia.

Response to reviewer: Thank you for your constructive feedback. We fully acknowledge the limitations of the ELISA assay used in our study, which detects antibodies against HEV but does not distinguish between HEV-A and HEV-C. We agree that this limitation precludes definitive conclusions about zoonotic HEV-C transmission. We have accordingly revised the manuscript to clarify that the human seroprevalence data likely reflect past exposure to HEV-A, not HEV-C. We have also tempered any statements regarding the zoonotic potential of HEV-C based solely on these data and now emphasize the need for future studies employing more specific serological or molecular tools to investigate potential HEV-C spillover. We have clarified methodological details and revised sections of the manuscript to improve structure and readability.

Materials and Methods:

The section of “IgG antibody detection in human serum” original reads: We tested participant sera for IgG antibodies against the three viruses of interest. Anti-HEV IgG antibodies were screened using a commercial HEV IgG enzyme-linked immunosorbent assay (ELISA) kit (Beijing Wantai Biological Pharmacy Enterprise, Beijing, China) according to manufacturer's instructions. Antiarenavirus and anti-hantavirus IgG antibodies using an in-house ELISA [6,27]. Positive samples identified during the primary screening were repeated. Presence of IgG antibodies was considered previous exposure to the pathogen. Seroconversion toward a virus was

defined as the transition from IgG seronegativity at baseline to seropositivity in the follow-up visit.”

Revised: For clarity, we revised this section as follow (lines: 250 – 265):

“We tested participant sera for IgG antibodies against mammarenavirus and Thailand orthohantaviruses using an in-house enzyme-linked immunosorbent assay (ELISA) (12,30). Due to the lack of HEV-C specific ELISA at the time of data analysis, we used a commercial anti-HEV IgG ELISA kit (Beijing Wantai Biological Pharmacy Enterprise, Beijing, China), which primarily detects anti-HEV-A IgG antibody, according to manufacturer's instructions. Although this assay does not differentiate antibodies against HEV-A and HEV-C, it was the only serological tool available in Cambodia.

Positive samples identified during the primary screening were repeated. Presence of IgG antibodies was considered previous exposure to the pathogen. However, we acknowledge that HEV seropositivity in our study may limit our ability to directly attribute results to HEV-C. Seroconversion toward a virus was defined as the transition from IgG seronegativity at baseline to seropositivity in the follow-up visit.”

Results: We have revised our result in the section **“Evidence of spillover of rodent-borne viruses in humans”** by adding the following statement to the HEV results (lines: 360 – 362): *“Due to the limited specificity of the Wantai ELISA assay, HEV seropositivity likely reflects prior exposure to HEV-A, rather than HEV-C, and therefore cannot be interpreted as evidence of zoonotic HEV-C transmission.”*

Discussion: We have also revised the section related to discussion of HEV-C transmission in human. The revised statement reads as follow (lines: 486 – 492): *“Interpretation of the HEV results requires caution, as the Wantai ELISA kit used cannot differentiate HEV-A from HEV-C (43). Although HEV-C RNA was detected in rodents, human serological findings likely reflects exposure to endemic HEV-A. We therefor cannot infer zoonotic transmission of HEV-C. Future studies using HEV-C-specific antigens or neutralization assays are needed to evaluate potential zoonotic transmission of HEV-C more accurately.”*

2. No sequence data or phylogenetic analysis is presented. This is a significant omission, as such data are critical to understanding the circulating virus strains in the region and their potential public health impact.

Response to reviewer: We sincerely thank the reviewer for this helpful comment. We have submitted the sequences generated in this study to NCBI GenBank, and performed phylogenetic analysis of the three viruses of interest.

In the Materials and Methods section (lines: 240 – 244), we revised the paragraph as follow: “*All nucleotide sequences generated in this study were submitted to GenBank and registered under accession numbers: PV818126 – PV818159 (mammarenavirus), PV805852 – PV805873, PV845596, PV845597, and PV871972 (orthohantavirus), PV805874 – PV805887 (HEV-C), and PV794171 – PV794299 (rodent COI gene).*”

Regarding the phylogenetic analysis, the results of the revised manuscript has been added to the Results section (lines 320 – 333) and as well as in figure 2. Consequently, the former Figure 2 has been renumbered as Figure 3.

Results (lines: 320 – 333): “*Sequencing was attempted for all positive samples. However, five could not be successfully sequenced for mammarenavirus, leaving 34 sequences for further analysis. Phylogenetic reconstruction demonstrated that 34 of mammarenavirus sequences obtained from Cambodian rodents clustered within lineage 4, aligning closely with sequences previously reported from the country (Figure 2A). This suggests a stable circulation of lineage 4 mammarenaviruses across multiple rodent hosts and environmental settings. For orthohantavirus, 24 of the 25 positive samples grouped with SEOV, a globally distributed pathogen commonly associated with *R. norvegicus* in urban environments. One additional sequence clustered with Thottapalayam orthohantavirus, historically linked to the Asian house shrew (*S. murinus*), underscoring potential for cross-species maintenance of this virus in peri-domestic settings (Figure 2B). Finally, all 14 HEV-C sequences clustered within genotype HEV-C1 (Figure 2C), with high nucleotide identity to previously reported strains from other countries in the region. This finding indicates a sustained presence of HEV-C1 in Cambodian rodent populations and supports the regional distribution of this genotype across Southeast Asia.*”

3. Please align terminology with ICTV standards. Specifically, replace "hantavirus" with "orthohantavirus" and "arenavirus" with "mammarenavirus" throughout the manuscript. Also, in line 23, "rat hepatitis E virus" should be used.

Response to reviewer: Duly noted. We have revised the manuscript accordingly.

4. The introduction should include background information on the abundance and distribution of rodent species in Cambodia. Additionally, a brief overview of the molecular virology of mammarenavirus, orthohantavirus, and HEV-C would provide useful context.

Response to reviewer: We sincerely thank the reviewer for this helpful and constructive comments. For clarity, we have extensively revised the introduction of our manuscript by incorporating reviewer comments on the abundance and distribution of rodent species in Cambodia as well as a brief overview of the molecular virology of the three viruses of interest.

5. In line 170, replace "XXXXXXXX" with actual GenBank accession numbers. Sequences should be properly submitted to NCBI.

Response to reviewer: Duly noted. We have submitted the nucleotide sequences to NCBI GenBank. The accession numbers have been stated in the revised the manuscript under section: **“Nucleotide sequence accession number” (lines 240 – 244)**, which reads as follow: “All nucleotide sequences generated in this study were submitted to GenBank and registered under accession numbers: PV818126 – PV818159 (mammarenavirus), PV805852 – PV805873, PV845596, PV845597, and PV871972 (orthohantavirus), PV805874 – PV805887 (HEV-C), and PV794171 – PV794299 (rodent COI gene).”

6. Line 173: As mentioned earlier, the ELISA kit used likely detects antibodies against HEV-A, not HEV-C. This should be clearly stated, and its implications discussed.

Response to reviewer: Thank you for your constructive feedback. We have carefully addressed this point. Please see our answers to comments #1.

7. Figure 1: The meaning of the color coding (blue, green, yellow) is unclear. Please include a legend to clarify.

Response to reviewer: For clarity, we have revised the legend of the figure 1 which reads as follow: “Figure 1. Study sites, rodent species, and virus prevalence in rodents. (A) Study site: The study was conducted in urban (Phnom Penh, depicted in purple), rural (Kampong Cham province, depicted in yellow), and in Semi-urban interfaces (Sihanoukville province, depicted in green). (B) Distribution of rodent species: The chart illustrates the overall distribution of rodent species identified in each interface. “Other” category includes: *Bandicota savilei* (n=1), *Rattus argentiventer* (n=2), *Mus sp.* (n=9), *Rattus sp.* (n=7). (C) Virus detection in rodents collected from each interface: The presences of mammarenavirus and orthohantavirus were screened using pooled organ samples (kidney, spleen, liver, and lung) from individual rodent. HEV-C was screened from individual rodent liver samples.”

8. Line 242: The claim of HEV-C spillover into humans is not supported by strong evidence. Detection of viral RNA in human specimens would be required to substantiate such a claim.

Response to reviewer: We are grateful to reviewer for raising this matter. For clarity, we have revised our statement regarding the serological results of HEV to avoid any confusion. Current lines statement (lines 357 – 362) reads as follow: “Serological evidence of exposure to rodent-borne viruses was observed in the study cohort, with seroprevalence of 12.7% (100/788) for mammarenavirus, 10.0% (79/788) for orthohantavirus, and 24.2% (191/788) for HEV among the total cohort. Due to the limited specificity of the Wantai ELISA assay, HEV seropositivity likely reflects prior exposure to HEV-A, rather than HEV-C, and therefore cannot be interpreted as of zoonotic HEV-C transmission.”

9. Line 135: The information regarding virus detection across rodent species would be better presented in a figure or table to improve clarity and reader comprehension.

Response to reviewer: We appreciate the suggestion to present the virus detection data across rodent species in a figure or table. After careful consideration, we believe that the current text presentation already conveys this information clearly and concisely, and that adding an additional figure or table would duplicate data already summarized in the Results (lines 320 – 330 and 341 – 349). To maintain a streamlined manuscript and avoid redundancy, we have retained the narrative format while ensuring that all relevant numbers and species details remain explicit for reader comprehension.

10. Line 284: The conclusion that HEV-C poses a significant public health risk in Cambodia appears overstated, given the lack of direct evidence of human infection.

Response to reviewer: We fully agree with the reviewer and we have revised the statement accordingly. The revised statement in the Discussion section (lines 404 – 405) reads as follow: “Rodent-borne viruses represent an important public health concern in Cambodia.”

11. Line 313: There is no introduction or background provided for HEV-A, despite its apparent relevance to the human serological findings.

Response to reviewer: We thank the reviewer for the comment. We clarify that HEV-C and HEVA are genetically distinct viruses. HEV-C is a rodent-borne virus, which is the focus of our study, whereas HEV-A primarily circulates among humans and is unrelated to HEV-C despite both being able to infect humans. Consequently, we did not provide any background on HEV-A in the Introduction, as it is outside the scope of this study.

In the Discussion section (lines 486 – 492), we have clarified that the human seropositivity detected using the commercial Wantai ELISA kit most likely reflects past exposure to HEV-A due to cross-reactivity, but this does not imply any direct link between human HEV-A infection and the rodent-borne HEV-C strains identified. This distinction ensures that readers can interpret the human serological results without conflating the two viruses.

12. The manuscript should include a discussion on the specific rodent species that were found to carry each virus, which is crucial for understanding transmission dynamics.

Response to reviewer: We thank the reviewer for this suggestion. The manuscript now clearly specifies which rodent species were found to carry each virus, as detailed in the Results section (lines 320 – 330 and 341 – 349). Mammarenavirus was predominantly detected in *Rattus exulans*, orthohantaviruses were mostly found in *R. norvegicus* (with single cases in *R. rattus* and *Suncus murinus*), and HEV-C was exclusively detected in *R. norvegicus*.

We also expanded the Discussion (lines 419 – 425) to interpret these host-virus associations, highlighting the potential roles of rodent ecology, species-specific susceptibility, and habitat preferences in shaping transmission dynamics.

13. The manuscript contains numerous grammatical and typographical errors. A thorough proofreading by a native English speaker is recommended.

Response to reviewer: Duly noted. Our manuscript has been proofread by a native English speaker.

Reviewer #2 (Comments for the Author):

Julia Guillebaud et al. investigated the prevalence and zoonotic potential of rodent-borne viruses in Cambodia by analyzing rodent and human samples collected over different time periods and geographic regions. While the study addresses an important public health issue and includes a broad collection of field samples, the manuscript remains largely descriptive and lacks essential virological analyses to support its conclusions on zoonotic risk.

Major Comments

1. The manuscript does not include phylogenetic analyses of the detected viruses. To evaluate zoonotic potential and evolutionary relationships, the authors should construct maximum likelihood (or equivalent) phylogenetic trees using the RT-PCR-derived sequences. This will allow readers to assess the genetic similarity between the rodent-derived viruses and known human-infecting strains. Without such foundational data, the zoonotic implications of the findings remain speculative.

Response to reviewer: We thank the reviewer for this thoughtful suggestion. The primary objective of our study was to assess the burden and distribution of rodent-borne viruses in Cambodia rather than to perform detailed evolutionary analyses. In response to the reviewer's concern, we constructed phylogenetic trees using the neighbor-joining method based on our RT-PCR-derived sequences from animal samples to illustrate the relationships among the rodent-derived viruses (new Figure 2). We chose the neighbor-joining approach because our focus was limited to depicting sequence clustering of the detected strains rather than inferring evolutionary rates or detailed divergence patterns.

We fully agree that maximum likelihood or other model-based approaches are optimal for evolutionary inference; however, given the scope of this surveillance study—which is centered on prevalence and burden—we consider the neighbor-joining analysis sufficient to demonstrate the genetic placement of our sequences within known viral lineages, since it is faster and requires less computational resource compared to the maximum likelihood model. We have clarified this rationale in the revised Methods sections under the heading “*Assessment of rodent-borne virus infection in animal samples and phylogenetic analysis*” (lines 217 – 228), which now read as follow: “*All PCR amplified fragments were sent for Sanger sequencing to a commercial sequencing facility (Macrogen, Inc., Seoul, South Korea) using the Big Dye Terminator v3.1*

Cycle Sequencing kit (Applied Biosystems). Chromatograms were sent back electronically to IPC for verification by visual inspection using CLC Genomics Workbench software (CLC bio, Cambridge, MA). Viral sequences were aligned with reference sequences of mammarenavirus, orthohantavirus, and HEV-C retrieved GenBank database using MAFFT v.7.490 (25). Phylogenetic trees were constructed using the Neighbor-Joining method with 1,000 bootstrap replicates, based on TN93+G models of nucleotide substitution, as recommended by the “Find Best DNA/Protein Model” tool in the MEGA 11 software (26). Trees were visualized and annotated using FigTree v.1.4.4 (27) and Inkscape 1.2 (<https://inkscape.org/>). Phylogenetic relationships were assessed using neighbor-joining trees as the study aimed to determine viral burden rather than perform detailed evolutionary analyses”

2. The authors should expand their discussion on zoonotic spillover potential. This includes analyzing whether the viral detections correlate with seasonality, geographic location, or specific ecological factors. Additionally, the authors should clarify if any human-adapted mutations were identified, or if human-to-human transmission has been documented or suggested for the detected viruses in other settings. Such information is critical for interpreting the public health relevance of their findings.

Response to reviewer: We thank the reviewer for this valuable suggestion. In the revised manuscript, we have expanded the Discussion to address the zoonotic spillover potential of the detected viruses, including correlations with seasonality, geographic distribution, and relevant ecological factors (lines 412 – 447).

With respect to the analysis of human-adapted mutations, we appreciate the importance of this topic. However, identifying specific adaptive mutations requires comprehensive genomic datasets and analyses that extend beyond the objectives of the present study, which focused on determining the burden and distribution of rodent-borne viruses in Cambodia. We have clarified this scope in the revised text.

Minor Comments

1. The reported trend of viral infection incidence across education levels, primary > no schooling > secondary or higher, does not suggest a clear dose-response relationship with education. The

authors should clarify the statistical and epidemiological significance of this pattern, and discuss whether other confounding factors may be involved.

Response to reviewer: We thank the reviewer for this insightful comment. We agree that the observed pattern of viral infection incidence across education levels (primary > no schooling > secondary or higher) does not show a clear dose–response relationship, and the associated p-values ($p = 0.03$ for HEV) are borderline. We have revised the manuscript to clarify that these findings should be interpreted with caution. We also acknowledge that other confounding factors, such as socioeconomic status, occupational exposure, and hygiene practices, may influence this association and warrant further investigation. This discussion has been added to the Discussion sections (lines 509 – 516), which reads as follows:

“Older age was a significant risk factor of higher HEV seroprevalence, as described in previous studies, reflecting cumulative lifetime exposure, while higher education levels appeared protective (44,45). Nevertheless, the association between seropositivity and education level did not show a clear dose-response relationship, and borderline p-value suggest caution; other confounding factors, such as socioeconomic status, occupational exposure, and hygiene practices, may influence this association.”

2. The seropositivity for arenavirus in humans shows marked variation. The manuscript should provide a more detailed interpretation of this fluctuation, considering biological factors (e.g., immune cross-reactivity), environmental exposures, or differences in assay sensitivity and specificity.

Response to reviewer: We thank the reviewer for highlighting this point. In the revised manuscript, we have provided more detailed discussion of the observed fluctuation in human mammarenavirus seropositivity. Lines 459 – 471 in the Discussion section currently read: *“The seropositivity of mammarenavirus fluctuated across the three interfaces, likely due to a combination of rodent ecological factors (density and species composition), environmental pressure (flooding, habitat disturbance), and human socio-behavioral differences (housing quality, food storage, and occupational exposure). In urban setting, dense population of rodent and human, constant exposure to contaminated food, poorly managed waste, and frequent*

*flooding of sewer systems likely explains the high initial antibody prevalence. However, once a large fraction of the urban population is immune, new seroconversions may slow, creating a plateau effect over time. In contrast, semi-urban setting may have lower rodent densities and moderately improved sanitation, resulting in the lowest antibody prevalence. Rural communities, despite only moderate baseline prevalence, experienced the highest incidence of new infections, probably driven by seasonal surges of *R. exulans* linked to rice harvesting, grain storage, and flood-related rodent displacement, as well as episodic high-contact events such as market visits or agricultural activities.”*

3. The dominance of certain viruses in specific rodent species suggests potential host-specific tropism or susceptibility. The authors are encouraged to present supporting data and elaborate on whether this pattern may be influenced by species biology, virus-host adaptation, or sampling bias.

Response: We thank the reviewer for this insightful comment. Our data indeed show that certain viruses are predominantly detected in specific rodent species—for example, mammarenavirus was mainly found in *Rattus exulans*, SEOV in *R. norvegicus* and *R. rattus*, and HEV-C in *R. norvegicus*. This pattern likely reflects a combination of factors:

1. **Host biology and behavior:** Species-specific traits such as habitat preference, social structure, and foraging behavior may influence exposure to viruses and facilitate intraspecies transmission.
2. **Virus–host adaptation:** Certain viruses may have evolved to replicate more efficiently in particular rodent hosts, reflecting host-specific tropism.
3. **Sampling bias:** While we aimed to capture representative samples across interfaces and species, differences in trapping success or habitat accessibility could influence apparent prevalence.

We have added a brief discussion of these factors in the revised manuscript (lines 419 – 425) to clarify the potential mechanisms underlying the observed host–virus associations.

Reviewer #3 (Comments for the Author):

The study documents the circulation of 3 families of zoonotic viruses in their rodent hosts and in humans in Cambodia, in more or less urbanized environments and in two different seasons.

Such studies are important to better understand the real risk to human health of circulating zoonotic agents in the rodent reservoir in the environments where they cohabit. This study is rigorously conducted and its limitations clearly identified by the authors.

However, I have several concerns.

1. The socio-epidemiological data gathered during the interviews are not clearly described. For example, are professions, animal handling, etc. collected? This would enable us to better understand which factors were tested in the subsequent analyses. Factors tested but not significant should be indicated in the results section.

Response to reviewer: We thank the reviewer for the comment. We have revised the Results section and added this relevant information. However, indicating factors that were not significant in the Results may not add substantial value, as the focus of the Results is to present the observed associations and key findings. Including non-significant factors could unnecessarily lengthen the section without improving clarity. We therefore prefer to report only the factors with statistically significant associations, as currently presented.

2. The authors use a pan-hantavirus RT-PCR, which does not allow precise identification of the virus in question (not all hantaviruses have been shown pathogenic to humans). Even if Seoul hantavirus is probable in rats, it would have been useful to specify it by sequencing or specific RT-PCR.

Response to reviewer: We appreciate the reviewer's careful reading of our methods. We would like to clarify that virus identification was indeed performed. All pan-hantavirus RT-PCR-positive samples underwent sequencing of the PCR amplicons, and these sequences were included in a comprehensive phylogenetic analysis. The results confirmed that 24 of 25 hantavirus-positive rodents carried Seoul orthohantavirus (SEOV), while one carried

Thottapalayam orthohantavirus. These data are presented in the Results and illustrated in Figure 2B. We have emphasized this procedure and the sequencing confirmation in the revised Methods to avoid any ambiguity. In the revised version, this analysis—now explicitly described in the revised Methods section—was applied to all three viruses investigated in this study.

3. Education is found to be associated with lower HEV seroprevalence only (line 269). Given that the mode of transmission and reservoir is the same as for the other two virus families tested, this result may seem surprising.

Response to reviewer: We agree that this finding is unexpected. As noted in the revised Discussion, the association between education level and HEV seropositivity did not show a clear dose–response relationship and was supported only by borderline p-values. We therefore interpret it with caution. It is plausible that unmeasured confounding factors—such as socioeconomic status, occupational exposure, or hygiene practices—may contribute to this association rather than education per se. We have highlighted these limitations and possible confounders in the revised manuscript (Discussion, lines 512 – 516) to clarify that the observed link should not be considered a robust or causal relationship.

4. Another point concerns the effect of climate on the observed detection rates, which is not discussed. Only Arenavirus are found to be more detected during the rainy season (line 297), can climatic conditions affect the persistence of the different viruses in the environment?

Response to reviewer: We appreciate this insightful comment. In the revised Discussion we note that mammarenavirus detection increased during the rainy season, particularly in semi-urban areas, likely reflecting higher rodent density and human–rodent contact following flooding. For orthohantavirus and HEV-C, however, no clear seasonal trend was observed. While climatic factors such as humidity and temperature can influence viral persistence, there is currently no clear evidence that the environmental stability of mammarenaviruses differs substantially from that of orthohantaviruses or HEV-C. Our data therefore suggest that the observed seasonality is more plausibly explained by ecological drivers—flood-related rodent habitat disruption and

increased opportunities for human exposure—rather than differential viral stability. This clarification has been added to the Discussion (lines 425 – 429).

The manuscript is concise; however, it would benefit from the addition of certain elements to better understand the results observed.

5. The pathogenicity of these viruses for humans is only briefly discussed (lines 86-87), and does not allow us to estimate the real burden on human health, especially in Cambodia (as stated in lines 284-285).

Response to reviewer: We thank the reviewer for this comment. We acknowledge that a detailed assessment of human pathogenicity and clinical burden is important; however, this was beyond the scope of our current study, which focused on determining the prevalence and distribution of rodent-borne viruses in rodents and evidence of exposure in humans. Estimating the clinical burden in humans would require dedicated epidemiological and clinical investigations, which could be a focus of future studies.

6. Information is lacking on the persistence of these viruses in their hosts in the introduction part. Are infections chronic or acute? This is important for interpreting infection dynamics over the lifetime of rodents. Lines 226-227: Were these rodents old?

Response to reviewer: We thank the reviewer for this suggestion. Rodent orthohantavirus and mammarenavirus infections are generally persistent/chronic, whereas HEV-C infections are typically acute but detectable for a limited period. We added this information in the Introduction to contextualize infection dynamics (lines 96 – 101). Approximated rodent ages were systematically recorded, and captured individuals were mostly adults, representative of the local populations. We clarified this in the Results sections (line 285).

7. Similarly, information on the persistence of antibodies detected in humans (lifelong? or just a few years?) is missing.

Response to reviewer: We thank the reviewer for this important point. As described in the followup results, we observed that a substantial proportion of participants maintained seropositivity over the two-year period: 36.4% for mammarenavirus, 89.5% for orthohantavirus, and 96.2% for HEV.

These findings indicate that antibodies against orthohantavirus and HEV are relatively long-lasting, whereas mammarenavirus antibodies may wane more rapidly. We have added this clarification in the revised Discussion (line: 517 – 522).

Other minor comments:

-Line 80: specify the genus of the Wenzhou virus

Response to reviewer: Duly noted. Genus information of the Wenzhou virus has been added (lines 116 – 117).

-Line 161: I'm not sure the term "speciation" is appropriate here. Molecular identification?

Response to reviewer: We thank for this comments. The term “speciation” is now replaced in line 230 by the term “Identification of rodents species”.

-Line 175: Precise if both anti-PUUV and anti-SEOV-like IgG antibodies were detected?

Response to reviewer: Duly noted.

-Line 216: precise "targeted virus of interest"

Response to reviewer: Duly noted

Re: Spectrum01394-25R1 (**Burden of rodent-borne viruses in rodents and zoonotic risk in human in Cambodia**)

Dear Dr. Janin NOUHIN:

Your manuscript has been accepted, and I am forwarding it to the ASM production staff for publication. Your paper will first be checked to make sure all elements meet the technical requirements. ASM staff will contact you if anything needs to be revised before copyediting and production can begin. Otherwise, you will be notified when your proofs are ready to be viewed.

Sincerely,
Peter Pelka
Editor
Microbiology Spectrum